# Optimization of Mechanical Properties of Cr_3_C_2_-Ni20Cr/Graphite Cold Sprayed Coatings

**DOI:** 10.3390/ma14133458

**Published:** 2021-06-22

**Authors:** Wojciech Żórawski, Anna Góral, Medard Makrenek, Dominika Soboń, Anna Trelka, Marek Bara

**Affiliations:** 1Faculty of Mechatronics and Mechanical Engineering, Kielce University of Technology, Tysiąclecia Państwa Polskiego 7, 25-314 Kielce, Poland; dsobon@tu.kielce.pl; 2Institute of Metallurgy and Materials Science, Polish Academy of Sciences, 30-059 Krakow, Poland; a.goral@imim.pl (A.G.); a.trelka@imim.pl (A.T.); 3Faculty of Management and Computer Modeling, Kielce University of Technology, 25-314 Kielce, Poland; fizmm@tu.kielce.pl; 4Faculty of Science and Technology, Institute of Materials Engineering, University of Silesia, 41-500 Chorzów, Poland; marek.bara@us.edu.pl

**Keywords:** Cr_3_C_2_-Ni20Cr/graphite coating, cold spraying, optimization, mechanical properties

## Abstract

This study analyzed the mechanical properties of cold-sprayed Cr_3_C_2_-25(Ni20Cr) blended with Ni-graphite as a solid lubricant deposited on 7075 aluminum alloy substrate. To optimize the coating properties, different sets of parameters (graphite content in feedstock, process gas composition, spraying distance, and traverse gun speed) were tested in the frame of the Taguchi experiment. The cold-sprayed coatings were evaluated for their chromium carbide and graphite content, hardness, and coefficient of friction. Analysis of the microstructure of the deposited coatings revealed that graphite as a soft and brittle component fills all voids in the coating and its quantity depends on its content in the feedstock. The experimental results show that the composition of the process gas has the greatest impact on the Cr_3_C_2_ content in the coating and the proportion of graphite in the sprayed blend directly affects its hardness. In the case of the coefficient of friction, the most significant parameters were the graphite content in the sprayed blend, the spraying distance, and process gas composition. The conducted verification experiment with the optimum parameter values allowed a coating with the highest hardness and the lowest coefficient of friction to be obtained.

## 1. Introduction

The emergence of new surface engineering technologies has always been associated with research on obtaining surface layers with better properties than those obtained with the methods used so far. Thermal spraying techniques provide wide deposition possibilities of such coatings on the cooperating surfaces of components. They allow the use of materials with significantly different physical and chemical properties, which makes it possible to obtain coatings with the required properties [1,2,3].

Cold spraying offers new opportunities in this area. The characteristic feature of the cold-spray process is the temperature of the gas stream, which is always below the melting point of the sprayed material. In this process, powder particles are injected into a high-velocity stream of the gas (300–1200 m/s) and accelerated by flowing through a converging—diverging nozzle toward the substrate. Upon impact, the metallic particles in the solid deform to the form of splats and create a coating. Because of the very high speed of the powder particles, a uniform coating with a small number of pores and very high adhesion and cohesion is formed [4,5,6]. At the current stage of development, this technology offers a great opportunity to obtain coatings with various properties. Firstly, the low temperature of the process avoids any phase changes in the feedstock powder, so the sprayed coating presents the same phase composition. Additionally, as other deleterious phenomena as particle oxidation, evaporation, melting, recrystallisation, or gas release are avoided, the obtained coatings are more durable with better bond strength. The research currently being conducted covers the deposition of composite coatings with a very diverse share of ingredients and include the matrix also containing a solid lubricant [7,8,9]. The basic problem is the selection of the matrix material and the solid lubricant and then determining their mutual proportions.

Graphite is widely used as a solid lubricant and is constantly being researched in thermally sprayed coatings [10,11,12,13]. Its use in composite coatings sprayed with cold gas, where the influence of heat is very limited, enables the formation of coatings with new properties. Additional possibilities are offered by cold spraying metal matrix composites. The interaction of particles of metal, ceramic, and graphite during the formation of a coating during cold spraying is a new challenge [14]. Soft and brittle graphite grains in the cold-spraying process must be protected when they impact on the substrate, therefore they are covered with a thin metallic layer. Chen et al. [15] used Cu-coated graphite grains to make an Al_2_O_3_ composite using low-pressure cold spraying. On the other hand, Huang et.al. [16] used Ni-coated graphite powder to obtain a composite whose matrix was aluminum. Moreover, in cold spraying, due to the lack of heat influence on the sprayed material, the spraying parameters are the basic issue, especially when depositing composite materials with a solid lubricant [17].

The solid lubricant content affects the hardness of the matrix and its bonding. Its high content in the composite is a source of readily available lubricant, which at the same time has a positive effect on the reduction of the friction coefficient. However, a high contribution of solid lubricant reduces the hardness of the composite and the cohesive force between the matrix particles, which in turn leads to a reduction in wear resistance. The selection of spray parameters can be the result of process modeling or the use of one of the experimental design (DOE) methods [1,18]. The Taguchi method is a tool used to optimize various processes in industry [19,20,21], including cold spraying [22,23,24]. In this paper, we experimented using the base powder Cr_3_C_2_-25(Ni20Cr), which has been used for many years to deposit wear-resistant coatings, mainly via HVOF spraying [25,26,27]. Several studies have also been carried out on the properties of these coatings deposited by means of cold spraying [28,29,30].

Unfortunately, there are no studies on the deposition of cermet coatings containing graphite with the use of high-pressure cold spraying. Obtaining such coatings requires the selection of an appropriate material for the matrix and a material with the properties of a solid lubricant. Therefore, high-pressure cold spraying optimization based on the Taguchi method of cermet Cr_3_C_2_-25(Ni20Cr)/Ni-graphite coatings containing graphite as a solid lubricant is a novelty. Moreover, the use of this optimization method will allow for the analysis of the influence of the composition of the coating material and the parameters of cold spraying on the content of Cr_3_C_2_ and graphite in the deposit microstructure, its hardness, and coefficient of friction.

The researchers of this study aimed to determine the influence of the composition of the coating material and the cold-spraying process parameters on the microstructure and mechanical properties of a coating with Cr_3_C_2_-Ni20Cr as a matrix and graphite as a solid lubricant.

## 2. Experimental Details

### 2.1. Sample Preparation and Characterization

Two commercial powders for preparing the spraying blend were used in this study: Cr_3_C_2_-Ni20Cr (Diamalloy 3004) (Oerlikon-Metco GmbH, Immelborn, Germany), the main composition, and Ni-25Graphite (Durabrade 2221) (Oerlikon-Metco GmbH, Immelborn, Germany), a solid lubricant (designated as Gr). The particle size distribution of the powder feedstock was measured with a HELOS H2398 laser diffractometer from Sympatec GmbH (Clausthal-Zellerfeld, Germany). Both powders were blended in a mixer in three different proportions (Table 1) for one hour to obtain a homogeneous mixture. The mixtures of powders were sprayed onto the Al 7075 alloy samples (400 × 30 × 5 mm) using the Impact Innovations 5/8 Cold Spray System (Impact-Innovations GmbH, Rattenkirchen, Germany) with the maximum output; *p* = 40 bar, T = 800 °C, and a Fanuc M-20iA robot (Fanuc Robotics Ltd., Oshino, Japan). Nitrogen, helium, and their mixture were used as the process gas (Table 1). Each coating was sprayed with a step size of 2 mm between 10 passes. The total range of cold spraying of one sample included four layers to ensure sufficiently thick coatings. Before spraying, samples were cleaned with acetone and grit-blasted using size 30 corundum. The microstructure and chemical composition of the powders and the coatings were analyzed via SEM (Jeol JSM-7100 (JEOL Ltd., Tokyo, Japan) and FEI XL 30 (FEI Company, Hillsboro, OR, USA) as well as TEM (FEI TECNAI G2 F20) (FEI Company, Hillsboro, OR, USA). For cross-section analysis, both powders and coatings were embedded in a resin and were successively polished by diamond suspensions of the gradations 3, 1, and 0.25 µm. The Cr_3_C_2_ and graphite percentage of each coating was measured using the ImageJ program (ImageJ 1.48, NIH and LOCI, Bethesda, MD, USA) based on six microstructure analyses. Phase composition was studied using a Bruker D8 Discover diffractometer (Bruker Ltd., Malvern, UK) with CoKα radiation. The Vickers hardness HV0.3 was measured five times on the cross-section of the deposited coatings with a microhardness tester from CSM Instruments (CSM Instruments, Peseux, Switzerland). Friction was investigated by using the T17 reciprocating slide tester (ITE, Radom, Poland). Three samples were tested in each case. The coating worked with the pin (50 HS steel, diameter 9 mm) without lubrication. The duration of the tests, speed, jump, and load were 0.5 h, 0.2 m/s, 40 mm, and 31.81 N, respectively.

### 2.2. Taguchi Experiment

To determine the influence of the composition of the sprayed powders and cold-spray parameters, nine coatings were deposited using the Taguchi method based on an L_9_ orthogonal array to reduce the number of experiments (Table 1). Based on initial experiments, the following tested parameters were changed: the weight content of Ni-graphite powder in the blend (A), gas composition (B), spraying distance (C), and traverse gun speed (D) (Table 2). The S/N coefficient was calculated for the following cases [22,23]:
(a).for “the larger, the better” features (e.g., Cr_3_C_2_ and graphite content in the coatings, hardness)
S/N = −10log [(1/n) Σ(1/y_i_^2^)](1)(b).for “the smaller, the better” features (e.g., the coefficient of friction)
S/N = −10log [(1/n) Σ(y_i_^2^)](2)
where: y_i_—the tested output factor, n—number of measurements per experiment.


## 3. Results and Discussion 

### 3.1. Powder Characterization

Cr_3_C_2_-25(Ni20Cr) powder is a mixture of chromium carbide grains (75 wt.%) and 25 wt.% of Ni20Cr alloy (Figure 1a). There is a clear difference in the grain morphology of both components of the mixture. The chromium carbide grains are characterized by an irregular shape with angular edges, while the Ni20Cr alloy grains have a spherical, oval, and elongated shape. It can be seen that the chromium carbide grains are larger than those of the Ni20Cr alloy. The metallographic examination of this powder (Figure 1b) shows that the Cr_3_C_2_ powder grains are porous, while the Ni20Cr alloy grains do not show any porosity or inclusions. The Cr_3_C_2_-25(Ni20Cr) powder mixture is characterized by the parameters d10 = 9.4, d50 = 24.7, d90 = 55.3 µm (Figure 2a).

The grain morphology of the Ni-25Graphite powder is shown in Figure 3a. Their irregular shape and sizes are visible. Based on the analysis of the cross-section of these grains (Figure 3b), it can be concluded that the graphite grains are completely covered with an even, thin nickel shell. Its thickness is in the range of 3–6 µm, but in some places it reaches even 10 µm (Figure 3c).

The particle size distribution measurement revealed significant grain size differences for both powders. The parameters of the Ni-25Graphite powder are d10 = 47.2, d50 = 80.8, d90 = 138.9 µm (Figure 2a), and for Diamalloy 3004 are d10 = 7.5, d50 = 19.9, d90 = 45.6 µm (Figure 2b). It follows that the Ni-25Graphite powder grains are much larger than the Cr_3_C_2_-25(Ni20Cr) powder grains.

Based on the phase analysis, it can be concluded that Diamalloy 3004 powder consists of a Cr_3_C_2_ phase (orthorhombic, a = 5.5329 Å, b = 2.82900 Å, c = 11.4719 Å, Space Group: Pnma (62)) and a Ni-Cr cubic phase with the lattice parameter (a = 3.547 Å) close to that of the Cr_0.25_Ni_0.75_ phase (cubic, a = 3.552 Å, Space Group: Fm-3m (225)) (Figure 4a). The phase analysis of the Ni-25Graphite powder showed that it consists only of pure nickel (cubic, a = 3.529 Å, Space Group: Fm-3m (225)) and graphite (hexagonal, a = 2.461 Å, c = 6.708 Å, Space Group: P63mc (186)) (Figure 4b).

### 3.2. Characterization of Cold-Sprayed Coatings

Figure 5 shows the cross-sections of the coatings sprayed in the Taguchi experiment. Four phases can be seen in each coating.

Their mutual distribution can be identified on the basis of the presence of elements in the coatings (Figure 6). 

The dominant light grey phase is the matrix of the coating, which consists of highly deformed Ni20Cr alloy grains. The dark grey phase is chromium carbide, the grains of which, when they hit the substrate at high speed, are crushed and then jammed in the substrate and the matrix [28,30,31]. It can be observed that the size of these grains is much smaller than in the sprayed powder. This is because they do not have the ability to deform and most of them crumble into smaller grains when hitting the substrate. The plastic Ni20Cr matrix enables the deposition of carbide grains in it. Such cracked and embedded Cr_3_C_2_ grains are clearly visible for each case of the coating spraying parameters of the Taguchi experiment (Figure 5). When a Cr_3_C_2_ grain impacts the Cr_3_C_2_ grain already embedded in the coating, it is crushed and reflected from the Cr_3_C_2_ grain surface, additionally causing its partial knockout from the Ni20Cr matrix. The black phase visible in the cross-sections is graphite, which is the third component of the coating. Graphite grains covered with a thin layer of nickel deform when they hit the substrate and they deposit. Graphite is a soft material and its grains are easily deformed due to the loose interlamellar coupling between sheets in the crystal structure. It can be assumed that the Ni20Cr and Cr_3_C_2_ alloy powders hitting the surface successively cause further deformation of the deposited Ni-graphite grains, breaking its nickel shield and, at the same time, closing the graphite in the resulting voids and removing it from the coating surface. During the spraying process of coatings with the addition of graphite, a significant increase in dustiness was visible. The presence of graphite is clearly visible in the cross-section of coatings deposited in the frame of the conducted Taguchi experiment. It is located both in the voids between the deformed Ni20Cr alloy grains, on the interface between Ni20Cr and Cr_3_C_2_, and inside the cracked chromium carbide grains.

At higher magnification, strongly deformed graphite grains coated with nickel are also visible (Figure 7a). The deformed nickel layer surrounds the graphite grain and, at the same time, closely adheres to the crushed Cr_3_C_2_ grains and the Ni20Cr matrix (Figure 7b). The hard grains of Cr_3_C_2_ and Ni20Cr resulted in a much greater fragmentation of graphite grains in the coatings than in the case of the cold-sprayed 40 vol.% Al/Ni-Gr coating. Some deformed graphite grains have been preserved in it, which can be assumed to be the result of the interaction of softer Al grains and much lower pressure and temperature of the process [17]. The presence of graphite grains is clearly visible on the carbon distribution map (Figure 6), where their even distribution throughout the coating is visible.

The coating sprayed with cold gas from the same powder (Diamalloy 3004) without the addition of graphite has a completely different distribution of carbon resulting from its presence in Cr_3_C_2_ [30]. The analysis of the microstructure of the coating at very high magnifications (Figure 8) confirmed that the graphite accurately fills all the spaces of unevenness in the coating (Figure 8). Due to its structure, graphite is resistant to compression, while its shear strength is low and therefore it is easily deformed. Moreover, it is clearly seen that, upon impact, the Cr_3_C_2_ powder grains also break into nanometric grains, which are embedded in a metallic Ni20Cr matrix (Figure 8).

The analysis of the coating microstructure made in TEM Bright Field mode confirmed that the graphite fills all spaces, even of a nanometric size, both in contact with the Ni20Cr matrix and the Cr_3_C_2_ grain (Figure 9a). The STEM microstructure also shows a very good connection of the Ni20Cr matrix with the Cr_3_C_2_ grain (Figure 9b).

The fourth phase in the coating, brighter than the Ni20Cr matrix, is nickel. It is clearly visible in the case of weakly deformed Ni-Gr grains (Figure 7a), where it still forms a graphite grain cover. Mostly, it occurs in the form of very thin lamellae with a thickness of 0.5 μm to several micrometers in the immediate vicinity of the graphite. Since nickel appears only as a thin layer on the surface of the graphite grains (Figure 3c), it does not undergo the typical plastic deformation as in the case of spheroidal metallic Ni20Cr grains. After depositing on the surface, the nickel-coated graphite grain is very strongly deformed by the impacting Ni20Cr and Cr_3_C_2_ grains. This breaks the nickel shell and removes some or all of the graphite from it. The thin nickel shell adapts very well to the unevenness of the resulting coating and is clearly visible in the form of bright and long stripes, which clearly differ in shape from the strongly deformed Ni20Cr alloy powder grains.

The nickel phase is most often present in the immediate vicinity of graphite (Figure 7b) but is also visible in the coating without graphite in its vicinity. All the described phases are very well connected with each other and in each case (Figure 5) they form a dense microstructure of the coating in which there is no porosity. Such a non-porous microstructure was also obtained by Huang et al. [16] for the low-pressure cold-sprayed 40 vol.% Al/Ni-Gr coating, although it was stated that some porosity would be required in the case of abradable coatings. The same effect was achieved by Chen et al. [32] spraying a Cu coating with 10 wt.% Al_2_O_3_ content and 5–20 wt.% copper-clad graphite content. The same coatings but without the addition of Al_2_O_3_ showed a few pores [15]. Therefore, the authors linked the elimination of porosity with the effect of mechanical hammering by Al_2_O_3_ molecules. In addition, Ling et al. [33], by low-pressure cold-spraying coatings with a mixture of Cu and Zn with the addition of 10–30 wt.% copper-coated graphite, received dense coatings without pores. The flame-sprayed coatings of nickel-coated graphite with 20–50 wt.% WC content were also pore free [12]. Figure 10 shows the surface morphology of all nine coatings sprayed in the frame of the Taguchi experiment. Cr_3_C_2_ grains embedded in the Ni20Cr matrix are clearly visible.

Some of the ceramic grains cracked when hitting the plastic substrate, and such cracked grains are found in the cross-sections of the coatings (Figure 5). Graphite is visible in the spaces between the deformed grains of the Ni20Cr powder and the chromium carbide grains. The addition of nickel-coated graphite grains in the sprayed mixture makes the surface free from pores present on the surface of coatings deposited without its addition [30,33]. For all sprayed coatings in the frame of the Taguchi experiment, quantitative image analysis was used to calculate the amount of reinforcing Cr_3_C_2_ carbide phase and the amount of graphite responsible for the frictional properties of the coating (Figure 11).

In each case, the amount of both Cr_3_C_2_ and graphite varied. As the graphite content in the powder mixture increased from 5 to 10 wt.%, the proportion of chromium carbide lowered. Figure 12 presents a phase analysis of the coatings sprayed with cold gas as part of the Taguchi experiment. No new phase appeared in the coatings and the phase composition of all coatings was the same as the composition of the powders from which the spray mixture was made (Figure 4). All coatings were sprayed at the maximum temperature achievable by the system of 800 °C at various parameters (Table 1). Neither the change of the distance (20–40 mm) nor the change of the gas mixture composition (N_2_, He) and the addition of graphite in the nickel shell resulted in the appearance of a new phase in the coating.

### 3.3. Analysis of the Taguchi Experiment for Cr_3_C_2_ Content in the Coatings

The results for Cr_3_C_2_ (vol.%), graphite (vol.%), hardness HV0.3, and friction coefficient tests and the obtained values of the S/N ratio in the frame of the performed Taguchi experiment are presented in Table 3. The analysis of the results revealed that the cold-sprayed coating in Experiment 3 exhibited the highest S/N ratio of 29.314 and the cold-sprayed coating in this experiment showed the highest chromium carbide content of 29.22 vol.%.

Table 4 shows the S/N response, as an average of the three obtained values with reference to Table 2, for the volumetric Cr_3_C_2_ content of the coating for each level of spray parameters. ΔM is the result of the value difference between the maximum and minimum S/N ratio for each spray parameter. Based on the Taguchi analysis, the higher the value of ΔM, the greater the influence of a given parameter on the analyzed quantity. The analysis of the results obtained in Table 4 showed that the highest value of ΔM (1.74) for the four tested parameters was related to the type of gas used.

This means that the type of gas used in the cold gas process has the greatest influence on the chromium carbide content in the coating. The remaining parameters, i.e., graphite content in the sprayed mixture, spraying distance, and speed, have a much smaller influence. The mean S/N response plot for Cr_3_C_2_ content in the coating is presented in Figure 13. There are significant differences between the changes in the S/N ratio when the spray parameters increase from the lowest to the highest level. The influence of the graphite content in the sprayed mixture is very small. The chromium carbide grains are deposited in the plastic Ni20Cr matrix, and soft and brittle graphite, regardless of its quantity, has a very limited influence on this process. During the spraying process, most of the deposited graphite is removed from the surface by the hard Cr_3_C_2_ and Ni20Cr grains, hence a significant decrease in its content in the coating in relation to the content in the sprayed blend (73.99 vol.%). However, a very large influence of the gas is visible, with the highest value of the S/N coefficient obtained for helium as a process gas. Helium is the gas that reaches a much higher speed in the cold-spraying process, which is related to its much lower density compared to nitrogen. The much higher speed also results in a higher speed of the sprayed powder grains. In the case of the Cr_3_C_2_ grains, which do not undergo deformation, their greater kinetic energy caused by the increased speed allows them to be embedded in a plastic Ni20Cr matrix. As shown in Figure 13, changes in the S/N ratio in the case of a change in distance are very small. It can be assumed that the decrease in the speed of the Cr_3_C_2_ grains in the case of spraying from a distance of 50 mm is small and therefore has no effect on the Cr_3_C_2_ content in the coating. On the other hand, it can be seen that the grains of the sprayed Cr_3_C_2_ powder at a distance of 30 mm are not affected by the bow-shock phenomenon [34]. A visible increase in the S/N ratio occurs with increasing traverse gun speed. For the highest value of the speed, the S/N ratio shows the highest value. It follows that as the traverse gun speed increases, the Cr_3_C_2_ grains less frequently hit the surface and more effectively settle in the substrate. At the same time, they are less exposed to the impact of successive Cr_3_C_2_ grains, which cause them to break out [35].

A statistical analysis of variance (ANOVA) was used to evaluate the significance of individual cold-spray parameters on the chromium carbide content in the coating. Based on the analysis of the results (Table 5), it can be concluded that the most important parameter influencing the chromium carbide content in the coating is gas composition, the contribution of which is 93.8%. The contributions of the other parameters, i.e., the content of graphite (1.2%), spraying distance (0.2%), and traverse gun speed (4.8%), are much smaller.

### 3.4. Analysis of the Taguchi Experiment for Graphite Content in the Coatings

Table 6 shows the volumetric content of graphite in the coatings sprayed in the Taguchi experiment. Graphite was also a tested parameter of the spraying process as a component of the powder blend. The analysis of the results showed that an increase in its content in the powder blend causes visible growth in its content in the coating (Figure 11). Graphite is a soft and brittle material; therefore, it can be easily removed by hitting it with much harder Cr_3_C_2_ and Ni20Cr grains. It can be assumed that the thin nickel shell is a protective barrier that limits its removal from the surface of the coating. The highest S/N ratio of 15.408 was obtained in Experiment 9 and this coating had the highest graphite content of 5.89 vol.%. It should be emphasized that its content inside the deposited coatings was much lower than in the sprayed blend (11.1 vol.%). The S/N ratio values for the graphite content in the coatings, calculated in the same way as for the Cr_3_C_2_ content in the coatings, are shown in Table 7.

It was predictable that the highest value of ΔM (13.31) would refer to a parameter related to the graphite content in the sprayed powder mixture. Therefore, it is obvious that the level of graphite content in the sprayed powder blend has the greatest impact on the graphite content in coatings among all the parameters examined. A similar relationship was obtained by Chen et al. [32], where an increase in the content of graphite powder covered with a copper shell from 5 to 20 wt.% in a mixture with Cu and Al_2_O_3_-10 wt.% caused a corresponding increase in the graphite content in low-pressure cold-spray coatings. The same effect was also obtained when spraying the same powders but without the Al_2_O_3_-10 wt.% addition. The obtained results also confirmed that the proportion of the graphite in the coatings was much smaller than in the sprayed blend [15]. Higher graphite content in the coating sprayed with the Al powder mixture containing 60 vol.% nickel-coated graphite on steel, reaching 15.5 vol.%, was obtained by Huang et al. [16]. The results of the analysis of variance for the graphite content in the coating are presented in Table 8. They confirm the earlier conclusion that the amount of graphite in the mixture of spraying powders has a major influence on the content of graphite embedded in the coating. The contribution of this parameter is dominant and amounts to 93%. The other parameters, i.e., gas composition, spraying distance, and traverse gun speed, whose contributions are 1.0%, 4.5%, and 1.5%, respectively, have a negligible impact.

Figure 14 shows the mean S/N response plot for the graphite content in the coating. The influence of the graphite content in the sprayed mixture is clearly visible and confirms that the mixture with the highest graphite content allows the coating with the highest graphite coating to be deposited. To a much lesser extent, an increase in the spray distance also has an effect on the graphite content inside the coating. It can be assumed that much larger graphite grains achieve lower speeds, which are further lowered by the increased spray distance. This leads to less deformation of graphite grains and keeping them in the coating. If the graphite grain hits the surface with high speed, then the nickel shell breaks and the graphite without this cover is exposed to impacts of hard Cr_3_C_2_ and Ni20Cr grains, which remove it from the surface of the coating.

### 3.5. Analysis of the Taguchi Experiment for Hardness of Cr_3_C_2_-25(Ni20Cr)/Ni-25 Graphite Coatings

The changes in hardness of the cold-sprayed coatings and the corresponding S/N ratio are presented in Table 9. The highest hardness of 492.2 HV0.3 was achieved by the coating sprayed in Experiment 2, for which the S/N ratio was 53.842. The mean S/N ratios for each level of the cold-spray parameters are included in Table 10. The biggest difference between the maximum and minimum S/N ratio ΔM (2.57), indicating the greatest impact of the parameters tested, was achieved for the graphite content in the sprayed feedstock. This result reflects the role of graphite in the microstructure of the coating. Despite the fact that its content in the sprayed coatings is much lower (1.10–5.89 vol.%) compared to the other ingredients, i.e., Cr_3_C_2_ and Ni20Cr, it has a significant impact on hardness. In the case of the sprayed coating with 10 wt.% Ni-graphite content, the hardness values are similar. A significant drop in hardness is especially visible in the case of coatings sprayed with a mixture containing 15 wt.% of Ni-graphite powder, where the graphite content inside the coatings is the highest.

The main effect plot of each process parameter on the coating hardness is shown in Figure 15. It can be seen that, as the content of graphite in the sprayed mixture increases, the hardness of the cold-sprayed coating decreases. Graphite is the most sensitive component of the coating. Its grains, when hitting the ground, behave quite differently than the other ingredients of the feedstock. Ni20Cr alloy grains undergo severe plastic deformation when they hit the surface of the coating and the associated cold working of already deposited layers, which in turn causes an increase in their hardness [36]. Hard and brittle Cr_3_C_2_ grains are not deformed but crushed during impact and deposition in the Ni20Cr matrix. Their presence increases the hardness of the coating. The graphite grains break easily, fill in unevenness and voids in the coating, and remain soft, reducing the hardness of the coating. Chen et al. [15,32] observed a similar relationship, the increase in the graphite content from 5 to 20 wt.% in sprayed powder also resulted in a significant decrease in the hardness of Cu/graphite and Cu/Al_2_O_3_/graphite cold-sprayed coatings. In general, the presence of graphite inside the metal matrix or metal—ceramic matrix reduces the hardness of the obtained composites, which is the result of its low hardness and poor interfacial bonding strength with the matrix components [11,15,32]. However, some authors reported an increase in hardness with an increase of the graphite content in sprayed blends [16,33,37]. A much smaller change in the S/N ratio in the case of gas composition is visible in Figure 15. Helium reaches a higher speed in the cold-spraying process, which causes an increase in the speed of the powder grains and, consequently, a more intense peening effect by hard particles. However, this phenomenon has little effect on the increase in coating hardness. The effect of spray distance (Figure 15) and traverse gun speed (Figure 15) on coating hardness is even smaller.

Table 11 presents the results of the analysis of variance for coating hardness. On the basis of the obtained results, it can be concluded that the graphite content in the feedstock is the most important factor affecting coating hardness, with a contribution of 89.2%. The gas composition, traverse gun speed, and spraying distance turned out to be insignificant factors, with contributions of 7.5%, 1.9%, and 1.5%, respectively. 

### 3.6. Analysis of the Taguchi Experiment for the Friction Coefficient of the Coatings

Table 12 contains the results of the coefficient of friction for the deposited coatings and the adequate S/N ratio. The lowest coefficient of friction of 0.499 was obtained in the frame of Experiment 6, for which the S/N ratio equaled 6.038. On the other hand, the highest coefficient of friction was 17% higher. The values of the average S/N ratio for each level of the cold-spray parameters are presented in Table 13. The greatest graphite content was in the sprayed blend for which there was the biggest difference between the maximum and minimum S/N ratio ΔM (0.77). It should be noted that the coating with the lowest coefficient of friction (0.499) contained relatively little graphite (1.64 vol.%). Moreover, the coating containing the most graphite (5.89 wt.%) did not achieve the lowest friction coefficient. This may also be due to the fact that two tested parameters, i.e., the spray distance and the composition of the processing gas, have a comparatively high impact on the coefficient of friction.

Figure 16 shows the effects of the cold-spray process parameters on the coefficient of friction of the deposited coatings. The results of the analysis of variance for the coefficient of friction are presented in Table 14. It follows that the magnitude of the coefficient of friction is also influenced by other tested parameters, not only the presence of graphite in the cold-sprayed coatings. Based on this analysis, the cold-sprayed coating should contain 5 vol.% graphite (Figure 16). In the reported tests, coatings with different graphite content were sprayed with the same parameters, which facilitated the analysis of its content on the properties of the deposited coatings [15,32,33]. The presence of graphite causes a reduction in the coefficient of friction due to its layered structure. However, its increasing amount in the coating is not associated with a further lowering of this coefficient. Based on tests of Cu-Al_2_O_3_ cold-sprayed coatings with the addition of 5, 10, and 20 wt.% graphite, Chen et al. [32] obtained the lowest coefficient of friction for its 10 wt.% content. In the case of Cu cold-sprayed coatings, the addition of 10, 15, and 20 wt.% graphite caused, in each case, almost the same reduction the coefficient of friction [15]. The same phenomenon, i.e., the same decrease in the coefficient of friction, regardless of the amount of graphite added, was observed in the case of cold-sprayed Cu-Zn coatings, where the addition of 20 and 30 wt.% graphite caused the same lowering of the coefficient of friction [33]. Another parameter that influences the coefficient of friction is the spray distance, which should take the lowest level, 20 mm (Figure 16). For this spray distance, the lowest coefficient of friction was obtained in Experiment 6. The third important parameter, slightly less than the previous two, is the process gas, and the use of helium also has a significant effect on the coefficient of friction. In both cases, both the lowest distance and the use of helium are associated with an increase in the velocity of the powder grains when they hit the substrate. Particle impact velocity is the main factor (apart from temperature) increasing the properties of the deposited coating [5,38]. It can therefore be assumed that the increased speed of all three components of the feedstock, i.e., Cr_3_C_2_, Ni20Cr, and graphite grains, resulting from the shortest spraying distance and the use of helium, reduces the coefficient of friction. The influence of the last parameter, gun speed, on the friction coefficient is negligible. Based on the analysis of variance (Table 14), it can be concluded that three parameters, the content of graphite, spraying distance, and gas composition, essentially affect the coefficient of friction and their contribution is 30.3%, 28.6%, and 23.4%, respectively. The contribution of the traverse gun speed is the smallest and amounts to 17.8%.

### 3.7. Verification Experiment

The obtained results indicate a diversified influence of parameters on the tested properties of the coatings. Graphite content in the sprayed blend had the strongest influence on the graphite content in the cold gas deposited coatings, its hardness, and coefficient of friction. However, in the first case, this parameter should have the maximum level (15 wt.% Ni-graphite), and in the other two, the minimum level (5 wt.%). The composition of the process gas had the greatest impact on the chromium carbide content in the coatings and on the coefficient of friction and hardness of the coatings. In either case, the use of helium gave the best results. Differences in the spray distance only affected the coefficient of friction, where the best effect was obtained at the shortest distance of 20 mm. For the last parameter, gun speed, its change affected the chromium carbide in the coating at the highest speed level. On the basis of the obtained results, a clearly differentiated influence of the graphite content in the sprayed blend on the tested coating properties is visible. Therefore, the optimum averaged values of the parameters for the properties tested have been calculated as the weighted mean separately for content graphite in the feedstock, gas composition, spraying distance, and gun speed using Equations (3) and (4):(3)$$w_{i}=\frac{\eta_{max}-\eta_{m}}{2\times SE}$$
(4)$$X_{i\left(opt\right)}^{\prime }=\frac{\sum_{i=1}^{n}w_{i}X_{i}}{\sum_{i=1}^{n}w_{i}}$$

The factor (*w_i_*) is the difference between the maximum S/N ratio (*η_max_*) and the mean S/N ratio (*η_m_*)—middle line on *η*-graphs, which was determined in relation to the standard error. 

On the basis of Equations (3) and (4), the values of optimal parameters were calculated; graphite content—10.5 vol.%, gas composition—15.5% N_2_ + 84.5% He, spraying distance—39 mm, gun speed—258 mm/s. A verification sample was then cold-sprayed using the above parameters. The conducted investigations showed that the coating was characterized by the following properties: the content of Cr_3_C_2_—27.2 ± 2.41 vol.%, the content of graphite—1.43 ± 0.31 vol.%, hardness—524.4 ± 42.68 HV0.3, and coefficient of friction—0.488 ± 0.032. The high content of chromium carbide seems to be the main factor determining its properties. The amount of particle reinforcement has a major influence on the hardness of the cold-sprayed coatings [37,38]. In this case, the coating sprayed with optimal parameters reached the highest values in the experiments carried out. The hardness of the Ni20Cr matrix, which is a predominant component of this coating, is also of great importance. Chromium carbide and Ni20Cr grains hitting the coating at high speed and the process gas consisting mainly of helium cause the overall tamping and work hardening effect, increasing its hardness. The high hardness is also the result of the low graphite content in the coating, which is a soft and brittle material and significantly affects hardness. The presence of solid lubricants reduces the hardness of the cold-sprayed composite coatings [7,12,15,32]. Therefore, it can be assumed that the bonds between all the components of the cold-sprayed Cr_3_C_2_-25(Ni20Cr)/Ni-graphite coating are strong enough to obtain the highest hardness. At the same time, the coating with such a low graphite content showed a lower coefficient of friction than that obtained in the experiments performed, which did not have to be intuitively obvious. The obtained coefficient of friction is the result of the joint action of the hard coating reinforcement and solid lubricant. In this case, it can be assumed that the content of hard reinforcement in the form of fine grains of Cr_3_C_2_ causes the decrease in the coefficient of friction as a result of lowering the contact area between the surface and the counter-sample by an increase in the hardness of the coating [37,38,39]. At the same time, even a relatively small presence of graphite as a solid lubricant in the coating causes an additional effect that reduces the coefficient of friction [17,39,40]. This mechanism is related to the formation of a lubricating tribofilm layer, which reduces the coefficient of friction. In general, both of the above mechanisms are related to the mutual arrangement of the components in the coating, i.e., the matrix, the reinforcing phase, and the solid lubricant, as well as the conditions of the experiment, so the optimization process seems necessary especially in this case [16,39].

## 4. Conclusions

The effect of graphite content in feedstock, process gas composition, spraying distance, and the gun speed on cold-gas-sprayed Cr_3_C_2_-25(Ni20Cr)/Ni-graphite coatings on aluminum alloy 7075 substrate were investigated with the use of the Taguchi experiment. The deposited coatings were characterized in terms of the chromium carbide and graphite content, hardness, and coefficient of friction. The main conclusions drawn are as follows:
The analysis of the microstructure of the coatings showed that graphite deposits in the high-pressure cold-spraying process, and it remains a component of the deposit and fills all the voids in it. Its content in the deposit is most significantly influenced by the graphite content in the sprayed blend. The highest graphite content in the coating, 5.89 vol.%, was obtained for the content of 15 wt.% Ni-graphite powder in the composite powder.Helium process gas has the highest impact on the Cr_3_C_2_ content in the coating and its use allowed for 29.22 ± 0.71 vol.% chromium carbide in the composite deposit.The graphite content in the sprayed mixture at the lowest level significantly influences the hardness of the coating. With the graphite content 1.21 ± 0.22 vol.% in the deposit, hardness reaches the highest value of 492.2 ± 41.9 HV0.3.The graphite content in the sprayed blend at a medium level—10.5 vol.%, the shortest spraying distance—39 mm, and a mixture of nitrogen—15.5 vol.% and helium—84.5 vol.% as the process gas have the greatest impact on the coefficient of friction. The cold-sprayed coating with the optimum averaged values of parameters revealed the highest hardness of 524.4 ± 42.68 HV0.3 and the lowest coefficient of friction of 0.488 ± 0.032.

## Figures and Tables

**Figure 1 materials-14-03458-f001:**
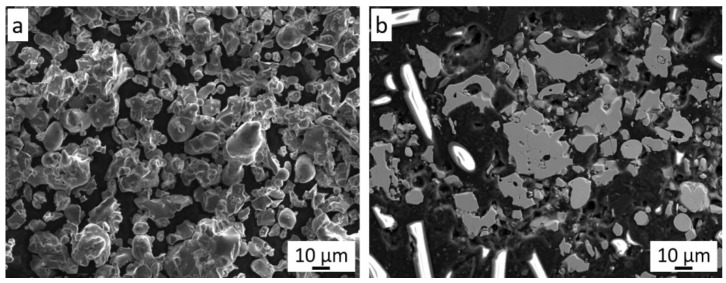
The Cr_3_C_2_-25(Ni20Cr) powder: (**a**) morphology, (**b**) cross-section.

**Figure 2 materials-14-03458-f002:**
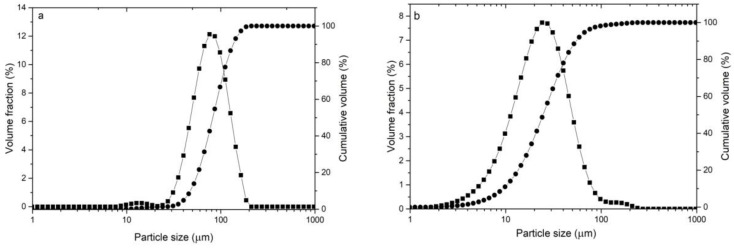
Grain size distribution of powders; (**a**) Ni-25Graphite, (**b**) Cr_3_C_2_-25(Ni20Cr).

**Figure 3 materials-14-03458-f003:**
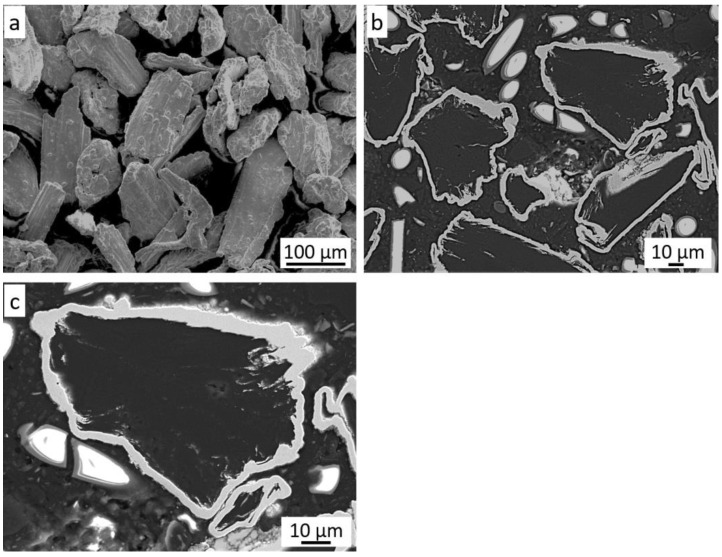
The Ni-25Graphite powder: (**a**) morphology, (**b**) cross-section, (**c**) nickel shell.

**Figure 4 materials-14-03458-f004:**
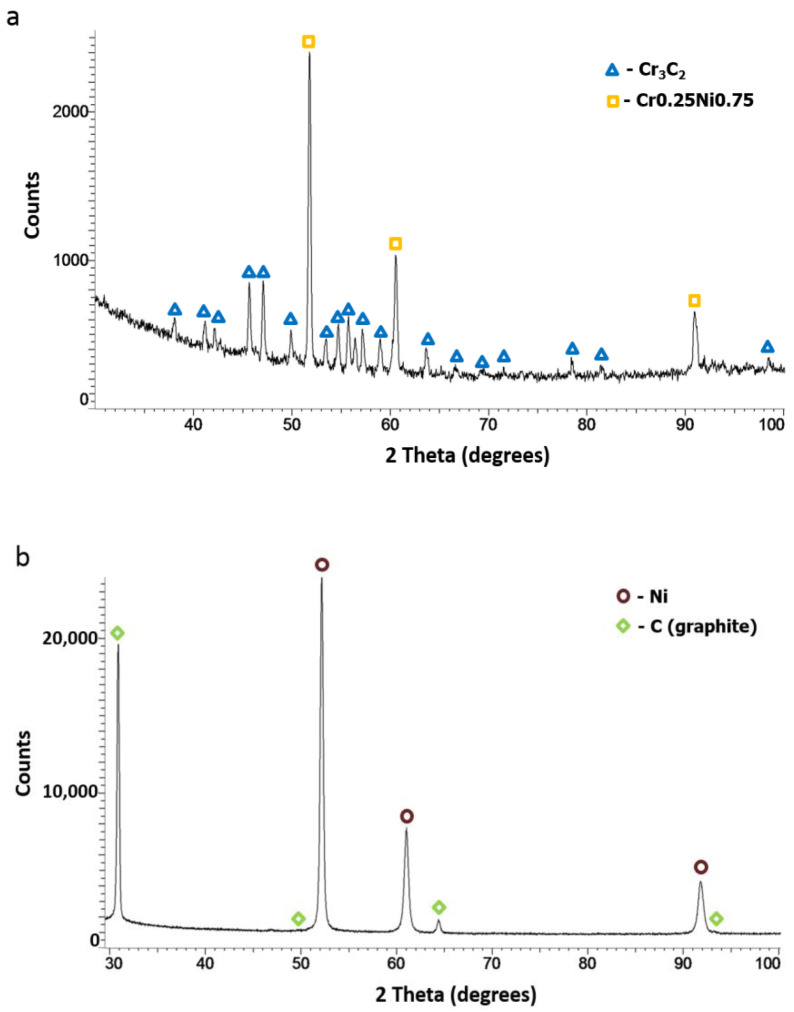
XRD patterns of powders: (**a**) Cr_3_C_2_-25(Ni20Cr), (**b**) Ni-25Graphite.

**Figure 5 materials-14-03458-f005:**
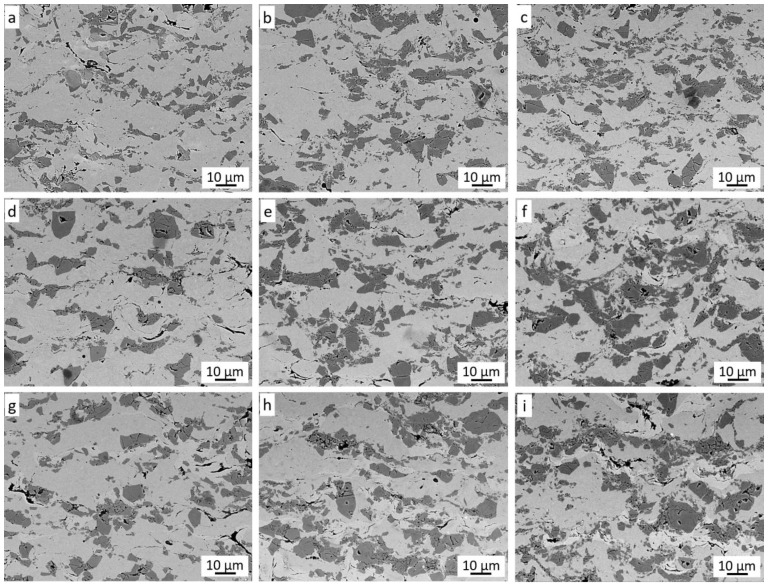
Cross-section of Cr_3_C_2_-25(Ni20Cr)/Ni-25Graphite coatings (**a**–**i**) deposited with the parameter sets from 1 to 9, respectively.

**Figure 6 materials-14-03458-f006:**
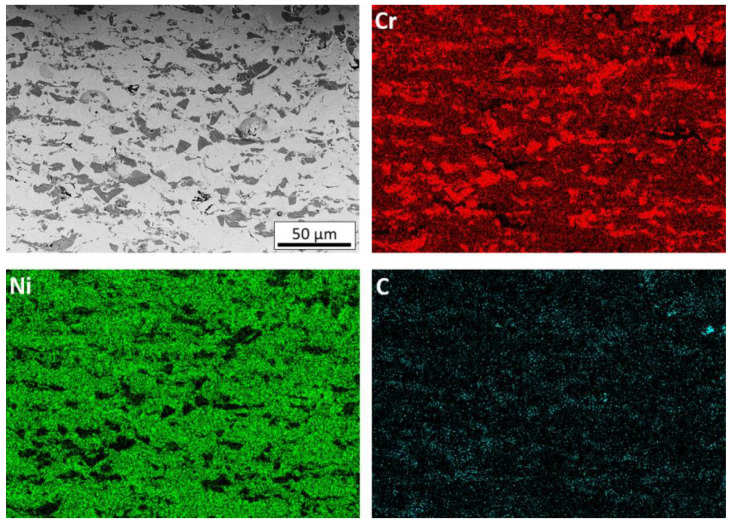
Cross-section and map of elements distribution (Cr, Ni, C) of the coating deposited with the parameter set 1 (SEM).

**Figure 7 materials-14-03458-f007:**
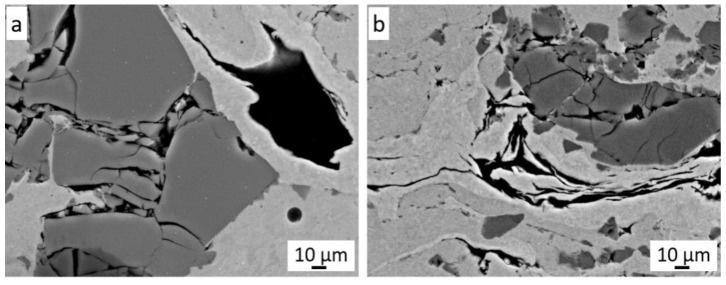
Slightly (**a**) and severely (**b**) deformed Ni-25Gr powder grains (black).

**Figure 8 materials-14-03458-f008:**
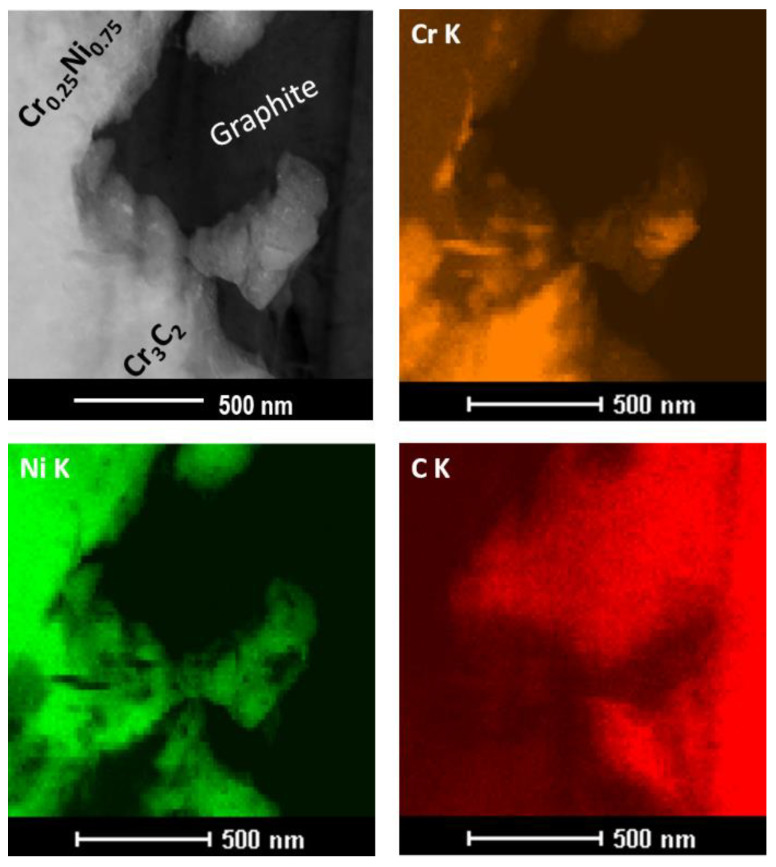
Cross-section and map of elements distribution (Cr, Ni, C) of coating deposited with parameters set 1 (TEM).

**Figure 9 materials-14-03458-f009:**
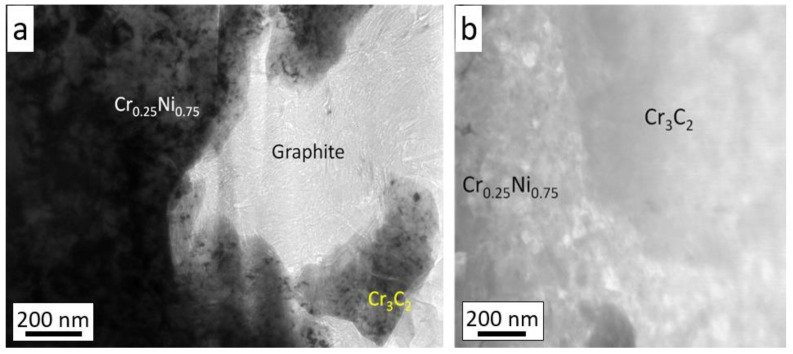
Cross-section of the Cr_3_C_2_-25(Ni20Cr)/Ni-25Graphite coating deposited with parameter set 1: (**a**) bright field mode, (**b**) STEM mode.

**Figure 10 materials-14-03458-f010:**
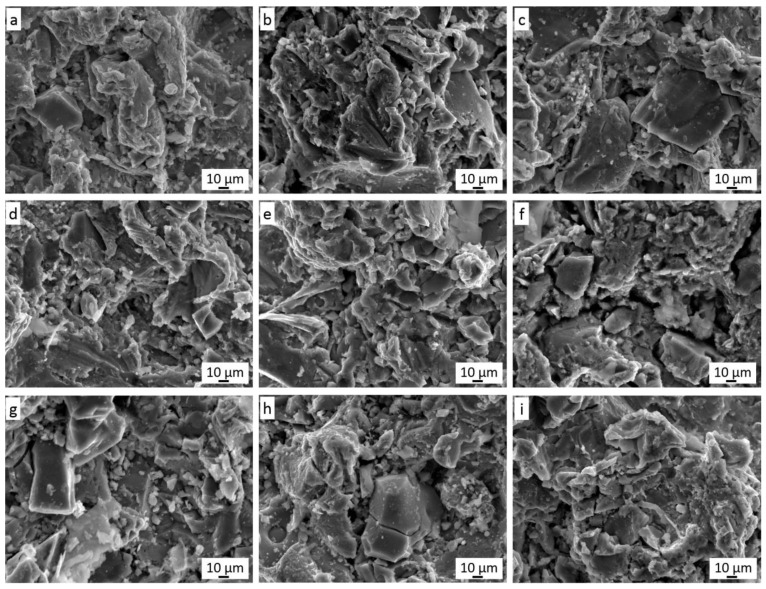
Surface morphology of Cr_3_C_2_-25(Ni20Cr)/Ni-25Graphite coatings (**a**–**i**) deposited with the parameter sets from 1 to 9, respectively.

**Figure 11 materials-14-03458-f011:**
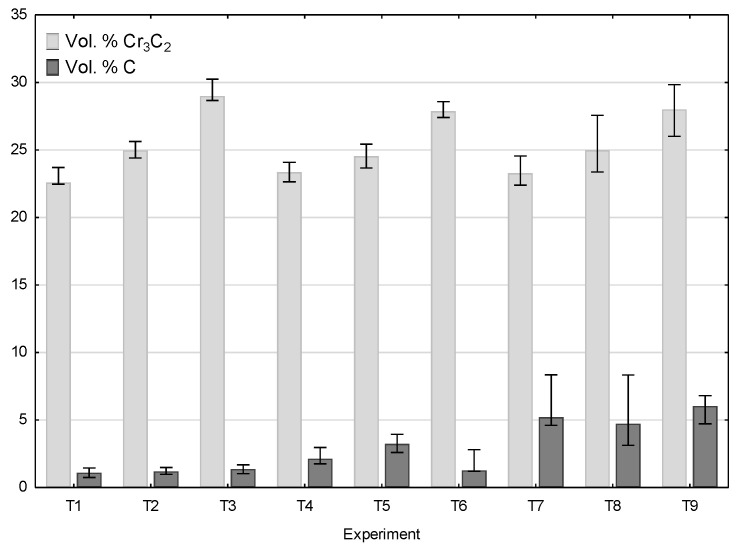
Chromium carbide and graphite volume content in Cr_3_C_2_-25(Ni20Cr)/Ni-25Graphite coatings deposited with the parameter sets from 1 to 9.

**Figure 12 materials-14-03458-f012:**
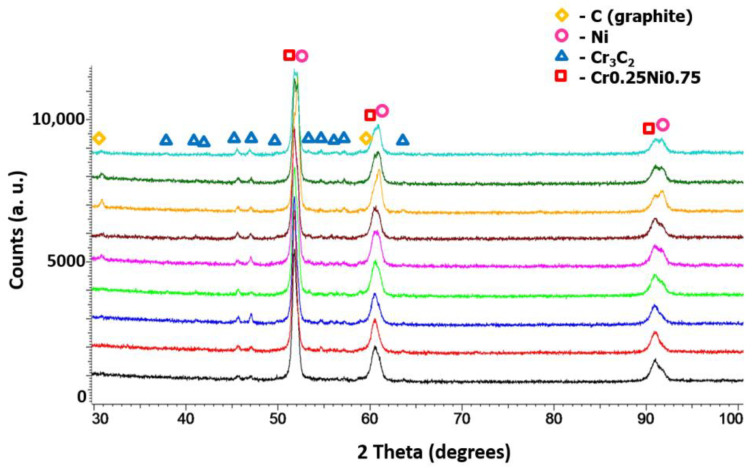
XRD patterns of Cr_3_C_2_-25(Ni20Cr)/Ni-25Graphite coatings deposited with the parameter sets from 1 to 9.

**Figure 13 materials-14-03458-f013:**
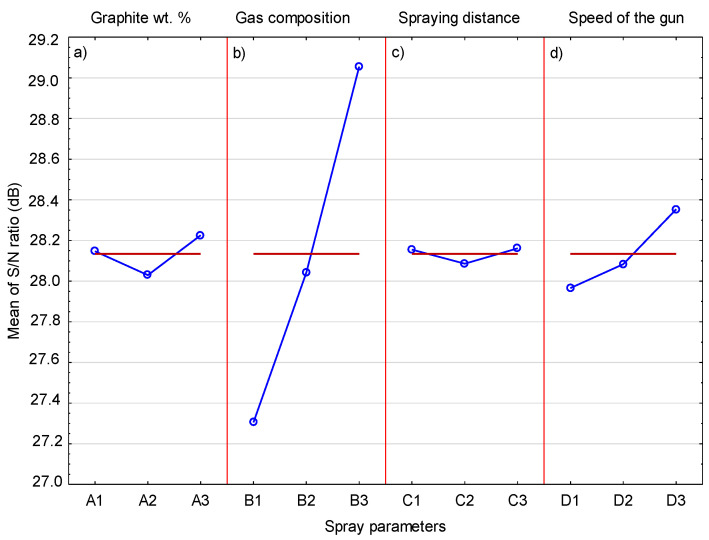
The mean S/N ratio for Cr_3_C_2_.

**Figure 14 materials-14-03458-f014:**
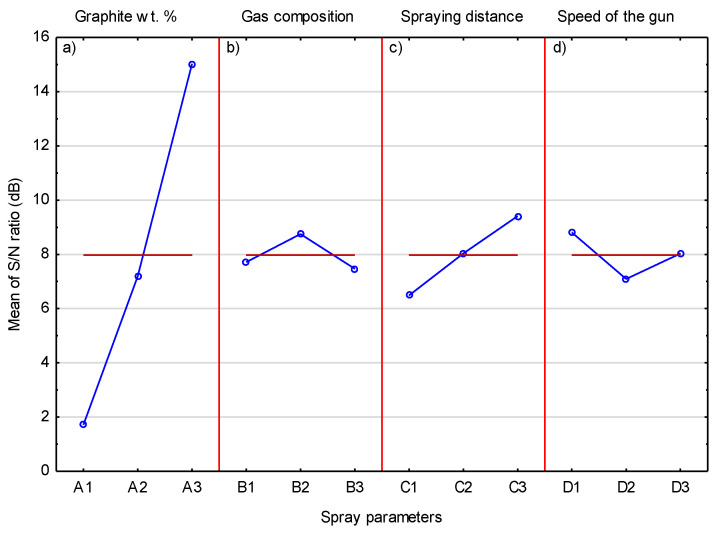
The mean S/N ratio for graphite.

**Figure 15 materials-14-03458-f015:**
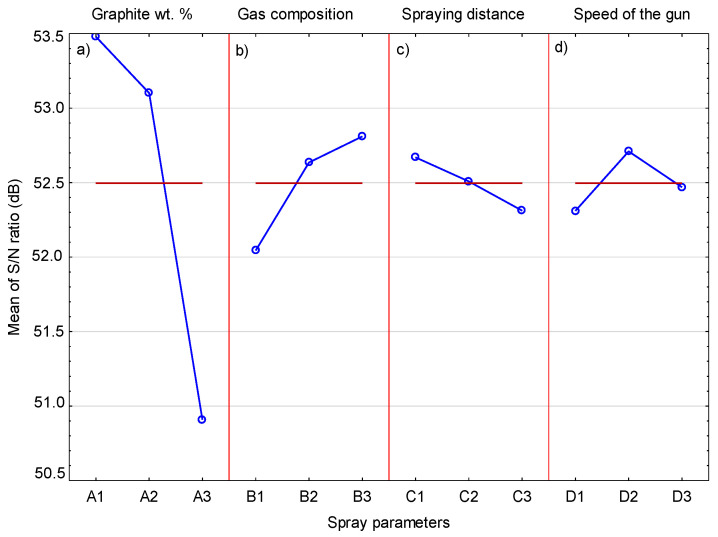
The mean S/N ratio for hardness.

**Figure 16 materials-14-03458-f016:**
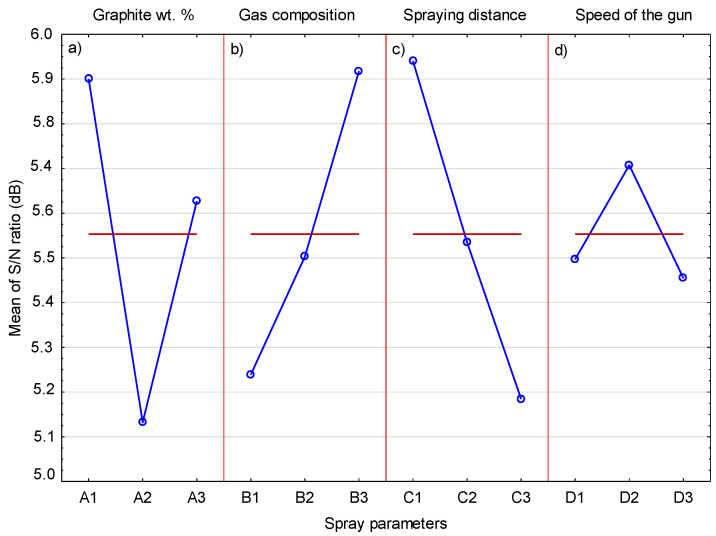
The mean S/N ratio for coefficient of friction.

**Table 1 materials-14-03458-t001:** Levels of tested parameters.

Level	Tested Parameter			
	Ni-Graphite Powder in the Blend, wt.% (A)	Gas Composition (B)	Spraying Distance, mm (C)	Traverse Speed of the Gun, mm/s (D)
1	5	N_2_	20	200
2	10	N_2_ + He	30	300
3	15	He	40	400

**Table 2 materials-14-03458-t002:** Taguchi orthogonal L_9_ (3^4^) design matrix.

Experiment	A	B	C	D
1	1	1	1	1
2	1	2	2	2
3	1	3	3	3
4	2	1	2	3
5	2	2	3	1
6	2	3	1	2
7	3	1	3	2
8	3	2	1	3
9	3	3	2	1

**Table 3 materials-14-03458-t003:** Results of the Taguchi experiment for Cr_3_C_2_ content.

Experiment No.	Cr_3_C_2_, vol.%	S/N for Cr_3_C_2_
1	22.83 ± 0.58	27.171
2	24.99 ± 0.57	27.956
3	29.22 ± 0.71	29.314
4	23.36 ± 0.59	27.371
5	24.54 ± 0.75	27.796
6	27.93 ± 0.53	28.920
7	23.37 ± 1.12	27.373
8	25.22 ± 1.73	28.372
9	27.96 ± 1.58	28.930

**Table 4 materials-14-03458-t004:** S/N response for Cr_3_C_2_.

Parameter	Mean S/N Ratio			
	Level 1	Level 2	Level 3	ΔM
A	28.15	28.03	28.23	0.20
B	27.31	28.04	29.05	1.74
C	28.15	28.09	28.16	0.08
D	27.97	28.08	28.35	0.39

**Table 5 materials-14-03458-t005:** Analysis of variance for Cr_3_C_2._

Parameter	SS	DOF	V	F-Ratio	P, %
A	0.0585	2	0.0293	5.5686	1.2
B	4.6292	2	2.3146	440.2972	93.8
C	0.0105	2	0.0053	7.8535	0.2
D	0.2361	2	0.1181	22.4590	4.8
Error	0.0105	2			
Total	4.9448	10			

SS—sum of squares, DOF—degree of freedom, V—variance, P, %—contribution of parameter.

**Table 6 materials-14-03458-t006:** Results of the Taguchi experiment for graphite content.

Experiment No.	Graphite, vol.%	S/N for Graphite
1	1.10 ± 0.31	0.796
2	1.21 ± 0.22	1.662
3	1.36 ± 0.29	2.696
4	2.25 ± 0.57	7.039
5	3.25 ± 0.57	10.226
6	1.64 ± 0.78	4.302
7	5.84 ± 1.69	15.308
8	5.22 ± 2.23	14.358
9	5.89 ± 0.88	15.408

**Table 7 materials-14-03458-t007:** S/N response for graphite content in the coating.

Parameter	Mean S/N Ratio			
	Level 1	Level 2	Level 3	ΔM
A	1.72	7.19	15.03	13.31
B	7.71	8.75	7.47	1.28
C	6.49	8.04	9.41	2.93
D	8.03	7.09	8.03	0.94

**Table 8 materials-14-03458-t008:** Analysis of variance for graphite content in the coating.

Parameter	SS	DOF	V	F-Ratio	P, %
A	268.3977	2	134.1989	20.88649	93.0
B	2.7704	2	1.3852	0.21559	1.0
C	12.8503	2	6.4252	0.29576	4.5
D	4.4462	2	2.2231	0.34600	1.5
Error	12.8503	2	6.4252		
Total	301.3149	10			

**Table 9 materials-14-03458-t009:** Results of the Taguchi experiment for hardness of Cr_3_C_2_-25(Ni20Cr)/Ni-25Graphite coatings.

Experiment No.	HV0.3	S/N for HV0.3
1	477.4 ± 42.1	53.013
2	492.2 ± 41.9	53.842
3	477.6 ± 16.0	53.581
4	428.2 ± 33.3	52.632
5	440.0 ± 28.8	52.869
6	489.8 ± 37.0	53.800
7	334.4 ± 31.7	50.485
8	362.8 ± 43.3	51.193
9	356.8 ± 85.7	51.048

**Table 10 materials-14-03458-t010:** S/N response for hardness of the coatings.

Parameter	Mean S/N Ratio			
	Level 1	Level 2	Level 3	ΔM
A	53.48	53.10	50.91	2.57
B	52.04	52.64	52.81	0.77
C	52.67	52.51	52.31	0.36
D	52.31	52.71	52.47	0.40

**Table 11 materials-14-03458-t011:** Analysis of variance for hardness of the coatings.

Parameter	SS	DOF	V	F-Ratio	P, %
A	11.5534	2	5.7767	60.13098	89.2
B	0.9667	2	0.4834	5.03112	7.5
C	0.1921	2	0.0961	1.13495	1.5
D	0.2422	2	0.1211	1.26034	1.9
Error	0.1921	2			
Total	13.1465	10			

**Table 12 materials-14-03458-t012:** Results of Taguchi experiment for friction coefficient tests.

Experiment No.	Coefficient of Friction	S/N for Coefficient of Friction
1	0.506 ± 0.006	5.917
2	0.502 ± 0.019	5.987
3	0.513 ± 0.004	5.798
4	0.582 ± 0.072	4.702
5	0.585 ± 0.039	4.657
6	0.499 ± 0.052	6.038
7	0.556 ± 0.016	5.099
8	0.509 ± 0.008	5.866
9	0.506 ± 0.005	5.917

**Table 13 materials-14-03458-t013:** S/N response for coefficient of friction.

Parameter	Mean S/N Ratio			
	Level 1	Level 2	Level 3	ΔM
A	5.90	5.13	5.63	0.77
B	5.24	5.50	5.92	0.68
C	5.94	5.53	5.18	0.76
D	5.50	5.71	5.45	0.25

**Table 14 materials-14-03458-t014:** Analysis of variance for coefficient of friction.

Parameter	SS	DOF	V	F-Ratio	P, %
A	11.5534	2	5.7767	60.13098	89.2
B	0.9667	2	0.4834	5.03112	7.5
C	0.1921	2	0.0961	1.13495	1.5
D	0.2422	2	0.1211	1.26034	1.9
Error	0.1921	2			
Total	13.1465	10			

## Data Availability

Data sharing not applicable.

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
