# Peer review of "Optimization of Mechanical Properties of Cr3C2-Ni20Cr/Graphite Cold Sprayed Coatings"

_materials, 2021, doi:10.3390/ma14133458_

Round 1

Reviewer 1 Report

Authors of this study have examined the mechanical properties of cold sprayed Cr3C2-25(Ni20Cr) blended in with Ni-graphite as a solid lubricant deposited on the 7075 aluminium alloy substrate. Their results of the the microstructure of the deposited coatings have revealed that graphite as a soft and brittle component fills all voids in the coating and its quantity depends on its content in the feedstock. Among other things, their experimental results have shown that the composition of the process gas has the greatest impact on the Cr3C2 content in the coating and the proportion of graphite in the sprayed blend directly affects its hardness. In fact, I found this work as interesting, with reasonably good pictures of publication quality. The discussions made as fundamental by nature. However, the technical quality of the paper is not well maintained.

I thus ask the authors the revise the paper based on the following suggestions:

  • Introduction is too tedious and the authors are advised to split it into several small paragraphs with clear motivation and objective.
  • Although lattice constants are reported, the authors have failed to report the geometrical details of the compounds examined. They thus should revise the second paragraph of page 5.
  • Background references are poorly presented. Authors should include the many important references that are missing in this study.

Reviewer 2 Report

Dear authors,

The manuscript materials-1247551, entitled ‘Optimization of Mechanical Properties of Cr3C2-Ni20Cr/Graphite Cold Sprayed Coatings’ presents the author approach on the optimization of properties of sprayed coatings onto aluminium substrate.

The deposition of the precursors (Cr3C2-Ni20Cr/Graphite powder) on aluminium substrate was performed by Taguchi method.

Different material characterization methods, such as: scanning electron microscopy (SEM: Jeol JSM–7100 and FEI XL 30), and transmission electron microscopy (TEM: FEI TECNAI G2 F20) were used for the characterization of Cr3C2-Ni20Cr/Graphite coatings. Particle size of the powder feedstock was measured via laser diffractometer (HELOS H2398 from Sympatec GmbH). Phase composition was studied using a Bruker D8 Discover diffractometer with CoKα radiation. The coating hardness was measured with a micro-hardness tester (CSM Instruments) and friction was investigated using the slide tester (T17 reciprocating, ITE Radom, Poland).

The results are presented well, supported by adequate images and graphs (with good resolution) revealing the infos needed to sustain the conclusions drown by the authors.

Overall, the manuscript is well written, with many techniques / methods described for the characterization of the sprayed films, with good correlation between the obtained results.

The conclusions are in brief, well organized and concise, sustained by the experimental findings.

However, small adjustments should be made on some figures, for better viewing, and understanding:

  • figure 4 on page 5 presents XRD patterns of powder needs to be made bigger, for better visualization;
  • I suggest also that the Figure6, on page 7, and fig 8 on page 8 to be made bigger, for better visualization, and with better contrast.

In the end, based on their experimental findings, the author concluded that the analysis of the microstructure of the coatings showed that graphite deposited and remains a component of the deposit and fills all the voids in it. The authors suggested that the parameter that has the greatest impact on the Cr3C2 content in the coating is the process gas, which should be helium. Moreover, they found that the hardness of the coating is significantly influenced by the graphite content in the sprayed mixture, which should be at the lowest level. Also, the cold gas sprayed coating with the optimum averaged values of parameters revealed the highest hardness and the lowest coefficient of friction.

However, I would expect that the conclusions to be more in detailed and with more perspective, and thus, I ask the authors to rewrite the conclusion (in the ‘expansion of the paragraph’ way).

I propose that this manuscript can be accounted for publishing after the above suggested MINOR REVISIONS in the journal MATERIALS.

Reviewer 3 Report

The paper deals with an interesting topic and is of possibly high interest to the readers. I have not found serious issues, and thereby I suggest only a minor revision. What I might suggest, though, is to specify the aim of the work better in the Introduction section, except that there are no studies on graphite containing cermet coating made with cold spraying. What is the real-life application of such coatings, where exactly would they be useful? Are the tribologic properties of such a coating expected to be superior to those prepared with conventional techniques?

Author Response

Please, the the attachment.

Reviewer 4 Report

The topic of manuscript is worth of investigation, and of interest for the readers. The authors performed many experiments and the manuscript is well organised.  Some papers could be added in the introduction:

International Journal of Molecular Sciences 22(6), (2021) 2917 

J. Am. Chem. Soc. 2015, 137, 14358–14366.

J. Electr. Eng. 2017, 68, 79–82.

I recommend the authors to improve the quality of the SEM  images (Figs. 1-2-5-7).

There are  no weak points in the manuscript. It could be published as the current form.

Round 2

Reviewer 1 Report

As expected, authors of the study have revised their paper based on my comments and suggestions. This is shown in the revised ms as red fonts. As such, the introduction, the results and discussion, and the conclusion sections have been significantly improved. I am now OK with this version of the ms. I therefore suggest publication of the research article in its current form.